

# Evaluating and analyzing the comprehensive community disaster reduction capability

Dajun Lian[1], Jin Zhu[1], Xinyu Wang[1], and Guangbin Li[1]

[1]Suzhou University of Science and Technology, Suzhou, Jiangsu 215009, China

*Correspondence to*: Dajun Lian (ldjwwyx@126.com)

**Abstract:** China currently faces the important task of strengthening its regional disaster reduction capabilities. Communities are basic components of urban areas, and ensuring that they participate in disaster reduction is important for urban safety. In this paper, in view of the imperfect evaluation criteria in the "National demonstration community of comprehensive disaster reduction (NDCCDR) ", and according to the connotation of community comprehensive disaster reduction capability (CCDRC), we construct an index system to evaluate the CCDRC; the system is comprised of six primary evaluation indices that measure the ability to evaluate disaster risk, the ability to provide rescue and support, the presence of engineering defenses, the presence of social and economic base support, the ability to manage disasters, and the level of public cognitive ability. These six primary indices include 31 secondary indices. Because the index system is characterized by a small sample size, sparse information, and large spatial extent, for the index system, we evaluate CCDRC using the entropy-weighted grey target model and geographic information system (GIS) overlay analysis. According to the distribution status of NDCCDR, we take the Suzhou New District (SND) as a case study for the empirical measurements and calculations and use ArcMap 10.2 software to produce a map of the spatial distribution of CCDRC in this region. The results indicate that the area's CCDRC is relatively weak. The spatial distribution of CCDRC is uneven. However, the CCDRC in the region has a good foundation and it also has large potential for improvement. The results also show that all of the NDCCDR are in the forefront of the case area, but their CCDRC is unbalanced and their primary evaluation indices of CCDRC are also not balanced. Therefore, we propose that the construction of NDCCDR and CCDRC should be combined, from point to face, and ultimately to improve the overall level of CCDRC in the community.

**Key words:** CCDRC; NDCCDR; entropy weighted grey target model; target center degree; GIS overlay analysis.

## 1. Introduction

Despite the capacity to understand and transform nature, human remain vulnerable to various natural disasters. Studies on the techniques, methods, and strategies involved in disaster prevention and mitigation remain important. In the past, engineering defenses have constituted the main approach to disaster mitigation. However, studies have suggested that this method is not always a successful means of disaster prevention and reduction (Gall et al., 2011; Deckers et al., 2010). Risk analysis can help decision makers identify high risk areas and thus use their limited capital and resources in the correct places (Hanne Glas et al., 2016; Shi et al., 2012). Therefore, risk analysis can significantly increase the effectiveness of disaster management, and it has become an active focus of research on disaster prevention and reduction in recent years. As the basic component of urban public disaster prevention and reduction strategies, communities play an important role in urban safety systems. More attention



should be paid to disaster risk evaluation at the community scale. *The national comprehensive disaster*
*prevention and reduction plan (2016-2020, China)* identified the "conduction of community disaster
risk identification and evaluation, and compilation of community disaster risk maps" as a major
strategy for disaster prevention and reduction over this time period (General Office of the State Council,
2017). Rapid urbanization and the interactions between various natural disasters mean that urban and
rural communities are impacted by many kinds of disasters, sometimes simultaneously. Therefore,
community-level disaster management has increasingly become the focus of global disaster
management (Li, 2012). Currently, the international community follows the concept of
"community-based disaster risk management" (CBDRM) (Bajet et al., 2008). Emphasis is placed on a
complete understanding of disaster risk (in addition to natural disasters, including public health, traffic
security, social safety, and accidents involving water, gas, and electricity) (UNISDR, 2012), a holistic
approach to disaster management (Zhan, 2006) and the universal characteristics of disaster prevention
and preparation (Zhou, 2013).

In recent years, China has experienced several major disasters, including the Sichuan (Wenchuan)
earthquake, the Yushu, Qinghai earthquake, and the Zhouqu debris flows in Gansu Province. Therefore,
the concept of comprehensive disaster reduction capability has gained significant traction. It is
generally recognized that comprehensive improvements in the disaster reduction capability will reduce
or mitigate the causalities and property loss caused by disasters (Hu, 2013). Prior to the International
Day for Disaster Reduction in 2006, the State Council of China convened the "Symposium on
enhancing the comprehensive disaster reduction capability", which focused on comprehensively
strengthening China's comprehensive disaster reduction capability. The concept of comprehensive
disaster reduction involves four main aspects: first, preparing for and defending against various kinds
of disasters; second, preparing to respond at different stages during the development of a disaster; third,
integrating various resources; and fourth, applying various disaster reduction approaches (Lyu, 2011).
Many studies have addressed the disaster reduction capability in China and other countries, and their
results have been adopted by the governmental organizations and committee and applied in actual
regional disaster management situations. These studies have mostly focused on single types of disasters
(Francesco Dottori et al., 2017; Zhang, 2004), single aspects of disaster reduction (Boris F et al., 2016;
Daniel Green et al., 2017), and the comprehensive regional disaster reduction capability (Ma, 2007).
Relatively few studies have addressed disaster reduction at the community scale. Although some
authors have constructed index systems for the ability of communities to prevent and mitigate disasters,
they did not propose an in-depth or specific quantitative method (Smith et al., 2017; Yi, 2012). In this
paper, we aim to fill this gap in knowledge by proposing a quantitative method of evaluating CCDRC.
We construct an index system for CCDRC. We quantitatively evaluate CCDRC using grey target
modeling and overlay analysis of GIS,and use the SND as a case study to demonstrate our calculations
referring to the distribution status of NDCCDR. We analyze the spatial distribution of CCDRC in the
study area with the goal of providing decision support for efficient disaster response management by
local government. The evaluation indices in this paper have the completeness, availability and
quantifiable characteristics. The model has the advantages of simple construction, space transferability,
simple operation, and multiple characteristics of evaluation results. Therefore, the method introduced in
this paper is universal.



## 2. Construction of an index system for evaluating the CCDRC

### 2.1 Defining the CCDRC

Based on CBDRM ideas, as well as the general philosophy of regional disaster reduction capability in China, we define the CCDRC as follows: a community's ability to avoid or reduce natural disasters and accidents involving public health, traffic security, and major utilities by using engineering and non-engineering measures to integrate resources from the government, non-governmental organizations, community residents, and the general public. These measures are taken during the process of disaster prevention and preparation, emergency response, and post-disaster recovery, with the aim of protecting the life and property of residents and supporting their normal activities (especially vulnerable groups), as well as the normal operation of industrial activities in the community. When evaluating the CCDRC, several aspects are critical: (1) while the disaster reduction capability at the community scale is an integral component of disaster reduction capability at the regional scale, they should not be evaluated or measured in the same way; (2) in addition to community organizations, the government, community, residents, and other organizations all contribute to community disaster reduction; (3) community disaster reduction is defined by the ability to cope with various kinds of disasters (both natural and man-made), not any single disaster; and (4) it is important to comprehensively consider various factors, including the evaluation criteria in the NDCCDR, when constructing an objective and comprehensive index system for evaluating the CCDRC.

### 2.2 Creating an index system for evaluating CCDRC

The NDCCDR uses demonstration as a means to enhance a community's ability to reduce disaster risk. The document defines ten aspects of organizational management mechanism, disaster risk evaluation, infrastructure of disaster prevention and mitigation and so on as the basic elements for compliance with the disaster reduction demonstration community (Office of National Disaster Reduction Committee, 2010). The wide-abroad implemented CBDRM attach importance to software construction but despise hardware environment construction. Compared with these disadvantages, the NDCCDR not only strengthen the planning and construction of software such as community residents' awareness and skills for disaster reduction, disaster reduction publicity and training, but also has taken into account the construction of community disaster reduction hardware such as shelters and material reserves. However, the above indicators are not enough to fully reflect the CCDRC. Therefore, based on the meaning of CCDRC, we consider quantitative factors including a community's economic status, rescue and safeguarding resources, and engineering defenses, as well as qualitative factors including disaster risk evaluation, organizational management, and public awareness of disaster prevention. Taking account of the universality, availability and quantifiable characteristics of indicators, we construct the evaluation index system of CCDRC. Our index system is made up of six primary indices and a total of 31 secondary indices, which mainly includes two types of attribute (spatial attribute and non spatial attribute), involve binary, numerical, and categorical data. The indices are listed in Table A1 of the Appendix A.

## 3. Evaluating the CCDRC

Our index system has the following characteristics: (1) it is hierarchical, but includes numerical,




binary, and categorical data; thus, these data cannot be processed in a standard way, and it is difficult to
determine their weights using traditional methods; (2) the index data do not have empirical values, and
the quantity of data is small, so quantitative evaluation is difficult; and (3) the same index differs
spatially, can evolve between communities and can be transformed to an index with consistent
polarization. Based on the aforementioned characteristics, we use grey target modeling to evaluate the
comprehensive disaster reduction capability of a single community. Next, we use GIS overlay analysis
to create a map showing the spatial distribution of the CCDRC throughout the region.

**3.1 Entropy weighted grey target model**

For evaluating data that involves a small sample and sparse and uncertain information, we first set
a grey target and take the bull's eye of the grey target as the standard model. The model is divided into
different grades based on the degree to which the model to be evaluated is close to the target center
(that is, the target center degree). This method is the traditional grey target model (Deng, 2002). We can
then consider the degree to which the various evaluation indices influence the target center degree and
use the entropy weight method to determine the weight of the evaluation indices, which will yield more
objective and fair evaluation results. This methodology constitutes the improved grey target model, or
the entropy-weighted grey target model (Li et al., 2012; Li et al., 2013; Li et al., 2016). The target
center degree of various spatial units can be calculated by the following Eq.(1):
$$\Gamma\left(X_0(j), X_i(j)\right) = \sum_{j=1}^{n} w_j \frac{minmin\Delta_{0i}(j) + 0.5maxmax\Delta_{0i}(j)}{\Delta_{0i}(j) + 0.5maxmax\Delta_{0i}(j)}, \tag{1}$$
The fraction on the right side of the equation is the target center coefficient of index $j$ ($j$=1, 2, …, $n$)
with a spatial unit of $i$; $\Delta_{0i}(j)$ is the corresponding grey correlation difference; and $w_j$ is the weight of
index $j$. Equation (2) shows the formula for calculating the entropy weight, as follows:
$$w_j = \left(1 - \overline{H}_j\right) / \left(n - \sum_{j=1}^{n} \overline{H}_j\right), \tag{2}$$
where $\overline{H}_j$ is the conditional entropy of index $j$ (Jin, 1994; Lian, 2004).

**3.2 GIS spatial overlay analysis**

Spatial overlaying is an important spatial analysis method in GIS. The method overlays two or
more image layers on the same scale in the same region to generate a new image layer with multiple
attributes. The new image layer synthesizes the attributes of the original image layers; this new layer
represents a new spatial relationship as well as indicating the relationship between the attributes of the
original image layers based on logical operations (Gong et al., 2001; Yang et al., 2016; He et al., 2015).
Overlaying a polygon image layer includes both intersection and superposition (Fig.1). Both the range
and attribute of a spatial unit will change after the intersection operation. This kind of overlay requires
logical operations and also includes complicated topological operations on spatial objects. After the
superposition operation, the range of the spatial units will not change, but the attributes will, mainly
because of the logical operations. When there many image layers are included in the operation, their
weights must be considered. In this paper, to determine a community's CCDRC, we take the
community as the evaluation unit and the image layers corresponding to various evaluation indices as
the objects of operation; we use the entropy-weighted grey target model for the superposition
operation.







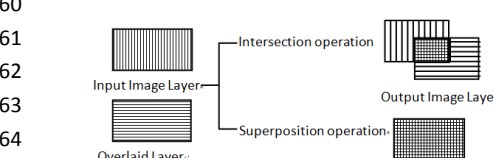

**Figure1. Schematic diagram of overlay operation.**

**3.3 Evaluation method and process**

Setting the community as the unit, we sequentially constructed the impact space of the index

sequence $\{U_j\}$ and the standard model $\{x_0(U_j)\}$ based on grey target modeling from the six primary
evaluation indices. The standard value of various indices in the impacting space is related to the index
polarity. The maximum standard value of index polarity is the maximum of the corresponding index in
the impacting space, and the minimum polarity index is the corresponding minimum. The specific
procedures of measurement and calculation are described below.

(1) Grey target transformation and determination of the grey correlation difference information: to

avoid the effect of large values (caused by excessively large differences between data values in the
standard model sequence) on the relatively small values, we conducted grey target transformation on
the various index sequences. After grey target transformation, the value of the evaluation index $U_j$ in
the i$^{th}$ research unit, $Tx_i(U_j)$, was calculated using the following Eq.(3):
$$Tx_i(U_j) = min\left(x_i(U_j), x_0(U_j)\right)/max\left(x_i(U_j), x_0(U_j)\right),\qquad(3)$$

Next, we obtained the grey correlation difference for index $U_j$ for unit $i$ in equation (1), $\Delta_{0i}(j)$, as

follows:
$$\Delta_{0i}(j) = |1 - Tx_i(U_j)|,\qquad(4)$$

(2) Calculation of conditional entropy $\overline{H}_j$ for index $U_j$: the relative distance between the index

$U_j$ in unit $i$ and the standard value $x_0(U_j)$ can be expressed with the closeness degree $d_{ij}$, as follows:
$$d_{ij} = 1 - \left[max\left(x_i(U_j), x_0(U_j)\right) - min\left(x_i(U_j), x_0(U_j)\right)\right]/\delta X_j,\qquad(5)$$
where $\delta X_j$ is the difference between the maximum and minimum of index $j$ for all the spatial units of
the study area. The normalized value of the uncertainty measurement (that is, the conditional entropy)
for the relative significance of this index can be expressed as follows:
$$\overline{H}_j = \frac{1}{lnm}\sum_{i=1}^{m}(d_{ij}/d_j)ln(d_{ij}/d_j),\qquad(6)$$
where $d_j = \sum_{i=1}^{m} d_{ij}$. If $d_{ij}=0$, we prescribed $(d_{ij}/d_j)ln(d_{ij}/d_j) = 0$.

(3) Calculation of the target center degree and grades for the various spatial units: first, we

calculated the entropy weight $w_j$ of the various indices $j$ using Eq.(2). Next, using Eq.(1), we calculated
the target center degree of the corresponding primary index of the different spatial units and the target
center degree of the corresponding CCDRC.

(4) Production of the CCDRC spatial distribution map: we import the above calculation results

into ArcGIS10.2, and the target layer of each level index of the study area will be set up. Then we carry
out the GIS overlay calculation according to the calculation method of the total target center degree,
and the CCDRC spatial distribution map of the study area can be obtained.



Using the CCDRC primary indices spatial distribution map and CCDRC spatial distribution map
generated in the case study area based on the above model, we can seek the following target: (1) the
overall level of CCDRC in case area, (2) the spatial distribution of CCDRC in the region, (3) the
potential analysis and improvement measures of CCDRC, (4) the CCDRC level of NDCCDR.

**4. Example calculations**

**4.1 The distribution status of NDCCDR**

Since the National Disaster Reduction Committee and Ministry of Civil Affairs of China
organized the selection of NDCCDR in 2008, nearly three thousand community have been selected (or
once been selected) in succession. Figure 2 (a) is the nationwide distribution map of NDCCDR in 2017.
The map shows that the selected communities are mainly distributed in the relatively developed capital
region (Beijing, Tianjin and Hebei, 117), the Yangtze River Delta (Jiangsu, Zhejiang, Shanghai, 241),
the Pearl River Delta (Guangdong, 126) and Shandong Province (92), accounting for 38.9% of the total
number. Figure 2 (b) shows that there are respectively 15 communities in Nanjing and Suzhou cities,
which are cities with the largest number of NDCCDR in Jiangsu province (115). The distribution status
shows that the NDCCDR construction of Suzhou is in the forefront of China;nevertheless,  the
construction of NDCCDR is a demonstration project of comprehensive disaster reduction work in
China, and it is also a component part of strengthening the comprehensive disaster reduction ability
across our country. Taking SND as an example, with 8 communities currently selected as the NDCCDR,
the NDCCDR construction work is obviously among the highest in China. However, the region has
jurisdiction over 82 communities, and the number of NDCCDR accounts for only 10% of the total
community. In addition, the evaluating index system of NDCCDR is imperfect compared with that of
CCDRC, and the NDCCDR is not necessarily consistent with the CCDRC. According to above two
aspects, we select SND as the research area to calculate and analyze the CCDRC. The characteristics
and distribution of the regional CCDRC can partially reflect the current situation and construction
direction of CCDRC in China.

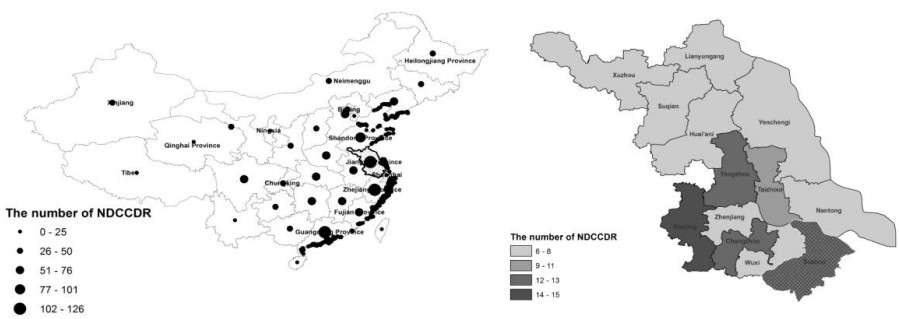

(a)  Distribution of NDCCDR in China        (b)  Distribution of NDCCDR in Jiangsu Province
**Figure 2. Distribution status of the NDCCDR in China.**

**4.2 Overview of the study area and data sources**

The city of Suzhou is located in the southeastern part of Jiangsu Province in China's Yangtze



Delta. It is a major part of the Yangtze River economic zone in Jiangsu. The area has a subtropical monsoon oceanic climate, with four seasons and abundant rainfall. In recent years, although large natural disasters have not occurred, various climate disasters have affected the day to day life of residents. Figure 3(a) shows a map of Suzhou. The study area is the SND, which is located west of the main urban area of Suzhou; it includes one major zone of economic development, three town level administrative districts, and four street administrative areas, totaling 83 communities. Figure 3(b) shows the administrative map. Since the 1990s, the SND has suffered from hailstorms, typhoons, freezing, and floods, which together have caused significant economic losses. In recent years, rapid economic development in Suzhou has resulted in a population boom and increased the frequency of man-made disasters. Thus, the local government has begun to focus more attention on enhancing the CCDRC in the area. Thus, eight communities, including Ylian, Hxiang, and Shshan, have been designated as NDCCDR (see shaded areas in Fig.3 (b)). These communities are mainly located in Xushuguan Town in the northeastern part of the study area and along Shishan Street in the southeastern study area. Thus, these communities have relatively strong organizations for responding to disasters and management capabilities, and they have focused on improving their ability to address disasters. However, CCDRC should also consider other aspects such as the community's rescue and support capability and engineering defenses. The model we present in this paper addresses the measurement and analysis of these aspects so that they can be strengthened.

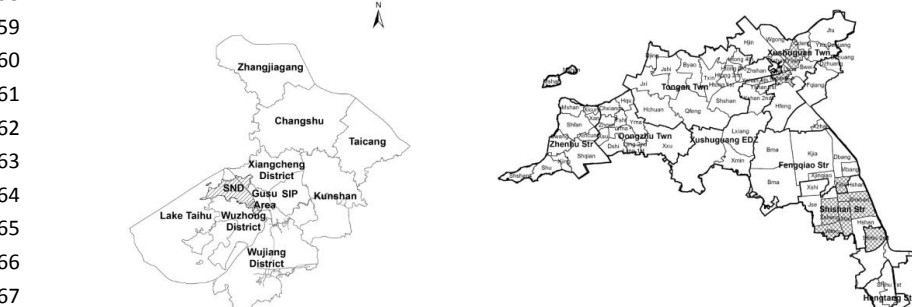

(a) Overview of Suzhou City.        (b) Administrative division of SND

**Figure3. Overview of research area and administrative district.**

In this paper, the data used for empirical calculation and analysis are from Suzhou City's spatial database, which includes the image layers for the districts, towns, and administrative boundaries including the communities, roads, water systems, and buildings (Fig. 4). The data were updated and examined before use. Table A1 of the Appendix A shows the indices of the non-building area ratio, the fortified area ratio of buildings, and the community road density, which were obtained using the statistical computation function of ArcGIS 10.2 software.

**4.3 Calculation of CCDRC in the region**

We surveyed and conducted statistical analyses on all 83 communities in the study area; we obtained effective sample data for 72 communities, which we used as the basic data. According to the grey target theory, we built the community-based influence space of the index sequence from six aspects, namely, disaster risk evaluation capability, rescue and support capability. In the light of





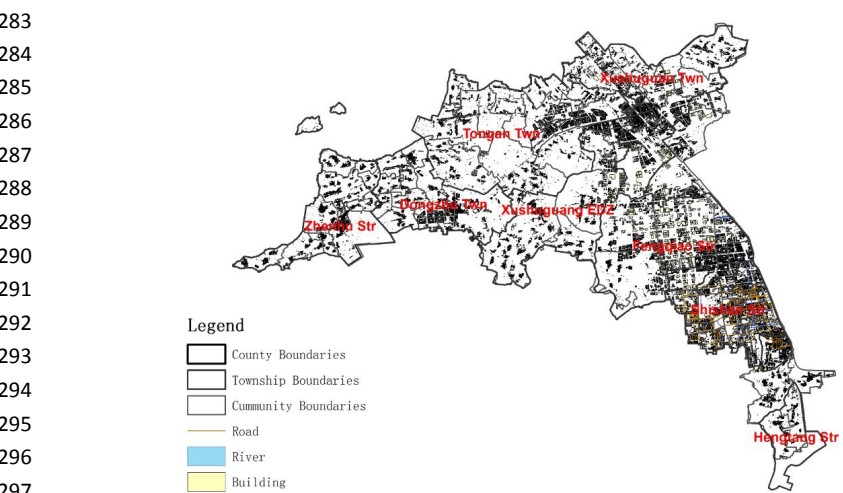

**Figure4. Image layer of spatial data in the study area.**

formula (1) - (6), we calculated the entropy weight, constructed the grey correlation difference information, and then obtained the target center degree. During the calculation, we need to pay attention to the index polarity. Except for number of group event disasters, number of fire disasters and the disaster risk intensity, all other secondary indices are maximum polarity indices.

We calculate the rescue and support capability for eight communities on Shishan Street as an example. Following the procedures above, we sequentially conducted grey target transformation, spatially determined the grey correlation difference information and conditional entropy, and calculated the entropy weight of each evaluation index (the calculated entropy weights for all the primary and secondary indices are shown in Table A1 of the Appendix A). Finally, we obtained the target center degree of the various primary evaluation indices. The results are shown in Table 1.

**Table 1.** Target center coefficients, entropy weights, and primary target center degrees for the rescue and support capability of communities on Shishan Street.

| Name of community / Evaluation index | Hshan | Jse | Shshan | Wfeng | Xsheng | Xtai | Hshan | Xshi | Entropy weight |
|---|---|---|---|---|---|---|---|---|---|
| Compilation of comprehensive asylum map | 1.00 | 1.00 | 1.00 | 1.00 | 1.00 | 1.00 | 1.00 | 1.00 | **0.045** |
| Disaster reduction capital investment (10,000 RMB/year) | 0.34 | 0.35 | 0.50 | 0.38 | 0.36 | 0.80 | 0.33 | 0.34 | **0.065** |
| Number of emergency rescue teams | 0.52 | 0.45 | 0.57 | 0.52 | 0.42 | 0.42 | 0.35 | 0.48 | **0.023** |
| Disaster information personnel (persons) | 0.34 | 0.34 | 0.34 | 0.34 | 0.34 | 0.35 | 0.34 | 0.34 | **0.059** |
| Reserve of rescue materials (10,000 RMB) | 0.34 | 0.35 | 0.37 | 0.36 | 0.34 | 0.36 | 0.35 | 0.37 | **0.036** |
| Per capita medical resources (/10,000 persons) | 0.33 | 0.33 | 0.33 | 1.00 | 1.00 | 0.35 | 0.33 | 0.34 | **0.061** |
| Target center degree | **0.454** | **0.451** | **0.495** | **0.602** | **0.588** | **0.554** | **0.440** | **0.454** | |

Note: the entropy weight corresponding to the various evaluation indices is the normalized weight determined with the 72 spatial units (communities) in the research area used as the reference.



The target center degree indicates the strength of the rescue and support capability. Table 1 shows
that, of the eight Shishan Street communities, the rescue and support capability is strongest in the four
NDCCDR, Wanfeng, Xinsheng, Xintai, and Shishan (Fig.3 (b)); it is relatively weak in the other
communities.
**4.4 Grading the CCDRC**
After inputting the data from Table 1 into ArcGIS 10.2, we conducted the GIS overlay operation
using the entropy-weighted gray target model described above, resulting in a map of the distribution of
the rescue and support capability in the Shishan Street communities. We repeated this operation for all
the communities in the study area and then graded their capabilities based on the target center degree.
We used anomalies to create four capability grades, excellent, good, moderate, and poor. Next, we
created a map of the distribution of the rescue and support capability grades for the communities in the
study area. We repeated the same operation for the disaster risk evaluation, engineering defense, social
and economic base support, disaster management and public cognitive capabilities of all the
communities, yielding the individual grade distribution maps for each capability. Finally, based on the
entropy weight of the primary evaluation indices (Table A1 of the Appendix), we used GIS overlay
analysis to determine the total target center degree of the CCDRC for all the communities in the study
area. Once divided into grades, based on the minimum information principle, the target center degree
should not be smaller than $1/(1+\zeta)$ (where $\zeta$ is the resolution coefficient, and its value in Eq. (1) is 0.5).
Therefore, the minimum of the total target center degree should be 0.3333. The results show that the
maximum total target center degree was 0.6941 for the Wanfeng community of Shishan Street.
Therefore, we created four grades in the interval [0.3333, 0.6941] and used these grades to create a map
of the spatial distribution of grades of CCDRC (Fig.5). The blank regions in the figure show
communities where we could not obtain qualified sample data.

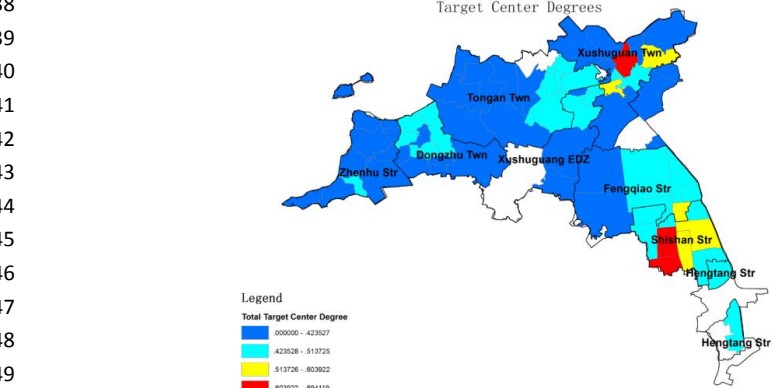

**Figure5. Spatial grade distribution map of CCDRC.**
**4.5 Result analysis and suggestion**
Comparing the spatial distribution of the grades of CCDRC in Fig.5 to the administrative
divisions shown in Fig. 3 (b), we observe the following characteristics:
(1) The CCDRC in the study area is generally weak. Communities with a poor CCDRC account
for 63.1% of the study area, and communities with a moderate grade account for 28.6%; communities



with an excellent grade only account for 8.3%. The distribution of these grades is not accidental but is
derived from the spatial distribution of the primary evaluation indices. For convenience of comparison,
we set the interval of the target center degree for all the primary evaluation indices to [0.3333, 1]. We
then divided the grades based on anomaly values to obtain the spatial grade distribution maps for the
primary indices (Fig.6). The figure is descending ordered from the upper left to the lower right based
on the entropy weight of the primary evaluation indices. It can be seen that the public cognitive
capability (Fig. 6 (a)) and the rescue and support capability (Fig.6 (b)), despite their maximum weight
values(0.381 and 0.288,respectively, visible from Table A1 of the Appendix ), show a general poor
feature, which is the main reason for the generally weak CCDRC in the study area.

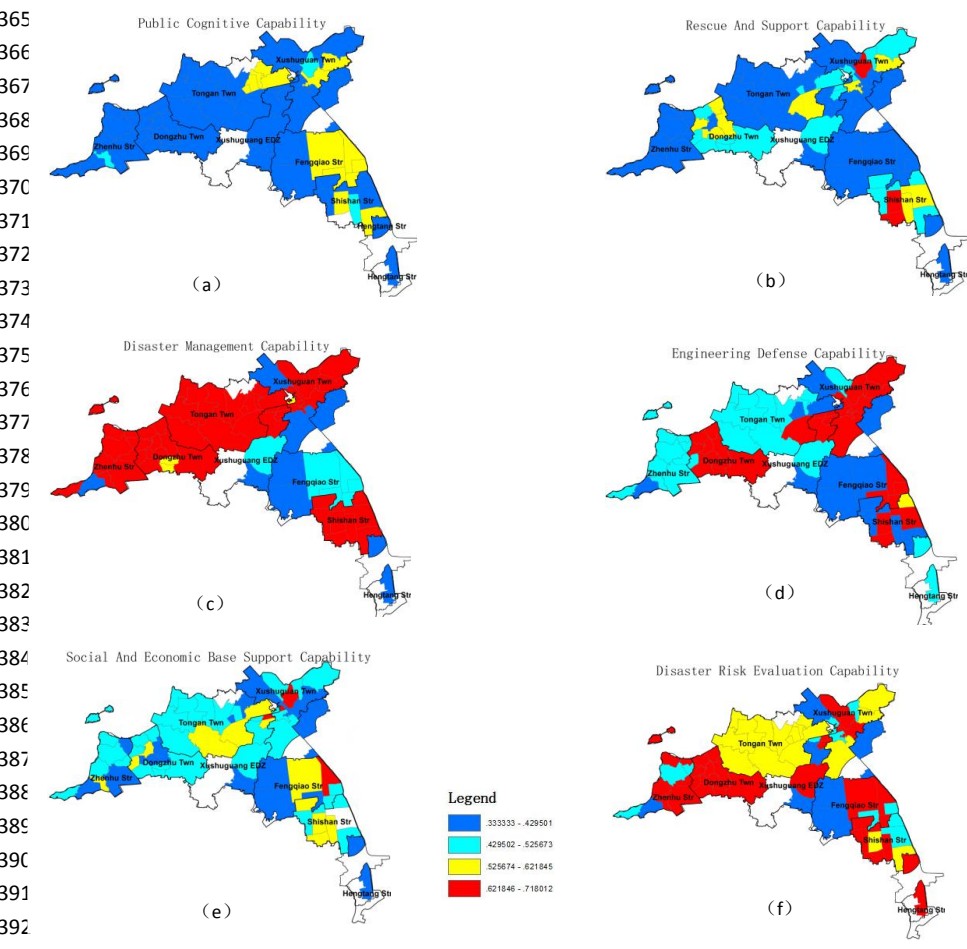

**Figure6. Spatial grade distribution map of various primary evaluation indexes in the community**

(2) The spatial distribution of the CCDRC is uneven. The eastern communities have relatively
greater CCDRC than the western communities adjacent to Lake Taihu. Communities in the
southeastern part of the research area generally have the strongest capabilities, and the CCDRC on
Shishan Street are the strongest. The communities of Xsheng and Wfeng have the best capabilities; the



community of Ylian, located in Xushuguan Town (in the northern part of the study area) has a grade of
excellent, and Hxiang and Yxin have grades of good. In contrast, the CCDRC is poor for most
communities on Zhenhu Street and in the towns of Dongzhu and Tongan, in the western research area
adjacent Lake Taihu. As mentioned in Sect. 4.2, all eight communities of NDCCDR in the research area
are located in Xushuguan Town and Shishan Street, and these communities selected as the 10 aspects
of organization management and disaster risk evaluation. Figures 6 (c) and (f) show that most of them
have excellent grades for disaster management capability and disaster risk evaluation; on the other
hand, their engineering defense capabilities (Fig.6 (d)) and social and economic base capabilities (Fig.6
(e)) are also in the overall advantage. Other communities, including Zhenhu Street (in the western
study area) and the town of Tongan (in the central part of the study area) have an excellent disaster
management capability, but their other primary evaluation indices are weaker. The uneven spatial
distribution of the disaster reduction capability is also related to the location. Shishan Street is
comprised of new urban villages constructed at the end of the twentieth century; it is bordered to the
east by the urban area of Suzhou (in the Gusu District), which has well-developed community facilities
and a high population. The Hxiang and Ylian communities are located in the central part of the town of
Xushuguan, which has a similar setting as Shishan Street. The towns of Tongan, Dongzhu, and Zhenhu,
in the western study area, are located on the edge of the Suzhou urban area. Most structures are houses
built by individual farmers or as part of settlement communities; living expenses are relatively low, and
there is a large transient population. The CCDRC in these communities is thus relatively weak.
(3) There is great potential to improve the CCDRC. The area ratio for the different grades of the
primary evaluation indices and CCDRC in the study area can be obtained from Fig. 5 and Fig. 6 (see
Table 2).
Table 2. Area ratio for the different grades of the primary evaluation indices and CCDRC in the study area
(unit: %).

| Grade | Public cognition | Rescue and support capability | Disaster management capability | Engineering defense capability | Economic base support | Disaster risk evaluation | CCDRC |
|---|---|---|---|---|---|---|---|
| Excellent | 1.2 | 3.1 | 63.0 | 33.8 | 4.7 | 39.3 | 3.1 |
| Good | 13.2 | 10.2 | 0.9 | 0.7 | 19.1 | 32.5 | 5.2 |
| Moderate | 3.2 | 21.5 | 12.3 | 35.5 | 46.0 | 8.9 | 28.6 |
| Poor | 82.4 | 65.2 | 23.9 | 30.0 | 30.1 | 19.3 | 63.1 |

Note: the primary evaluation indices in the table are shown in descending order from left to right based on their entropy weights.
The results show that communities with moderate or higher CCDRC account for 36.9% of the
study area; with the exception of the public cognitive capability and the rescue and support capability
(with the largest weights), communities with a grade of moderate or higher for other indices account
for over 76.1% of the study area, and communities with an excellent disaster management capability
(with the third highest entropy weight) account for 63%. To further analyze the potential for
improvements in the CCDRC, we use Tongan, which has a moderate CCDRC, as an example. Most
secondary evaluation indices of its engineering defense and community social and economic base
support capabilities are close to or better than the average level of the study area (as shown in Fig. 7,
where the vertical axis is the ratio between an evaluation index and the index's average value in the
study area). The analysis above indicates that there is a relatively large potential for improving the
CCDRC in the research area.
Above analyses suggest that we should focus on the following several aspects to enhance the
CCDRC in the study area:
(1) Several measures should be taken to improve the cognitive level in the communities. Because



its weight is largest, the public cognitive capability significantly affects the CCDRC. Figure 6(a) and
Table 2 both demonstrate that the public cognitive capability in the research area is insufficient,
communities with a poor grade account for 82.4% of the study area. In modern communities with
highly developed means of communication, many measures can be used to improve the public
cognitive capability. For example, some measures include creating official ways to disseminate disaster
reduction information, developing publicity material, setting up disaster early-warning display screens,
and increasing the amount of publicity material to increase residents' knowledge of disaster prevention
and reduction. Better publicizing disaster reduction activities will help residents understand the dangers
of disasters and instill a common sense of proper emergency behavior. Enhancing residents'
consciousness regarding disaster prevention and reduction will also help attract volunteers to join
disaster prevention and reduction teams and eventually strengthen the overall cognitive capability of
the public in the study area.

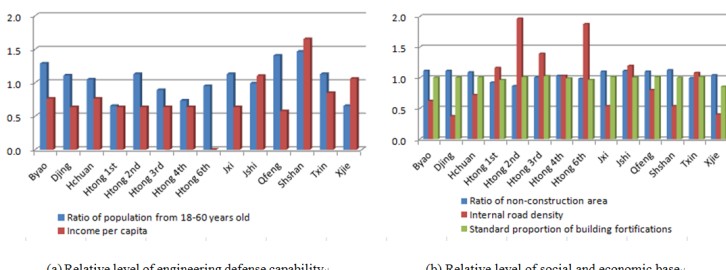

(a)Relative level of engineering defense capability            (b) Relative level of social and economic base

**Figure7. Relative level of engineering defense and social economic base support capability in Tongan Town.**

(2) It is also important to collectively manage and reinforce the effectiveness of disaster relief
measures and safeguards. Because the disaster management capability of communities in the study area
is relatively good (Fig.6(c)), we compare it to the target center degree of the community rescue and
support capability to yield a plot of the target center degree for the community rescue and support
capability (Fig. 8). This index is close to the minimum of the target center degree (0.333), in contrast to
the disaster management capability. We conclude that it is important to reinforce the disaster rescue and
support capability, including strengthening coordination between the relevant governmental
departments, investing in multiple aspects of disaster reduction and allocation of per capita medical
resources, appointing disaster information personnel, and setting aside more rescue and emergency
material. These measures will fundamentally strengthen the disaster rescue and support capability in
the study area.

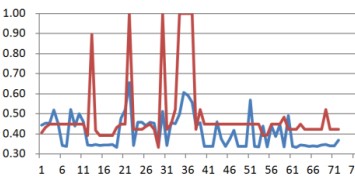

**Figure8. Curve for the target center degree of rescue and support capability**

(3) Similarly, the engineering disaster defense capability should also be strengthened. As shown in
Figure 6(d) and Table 2, the engineering defense capability in the study area is a bimodal distribution.
Communities with an excellent grade (including Dongzhu, Shishan Street, and some of Xushuguan)
account for 33.8% of the study area, while communities with a moderate or poor grade account for 35.5%




and 30%, respectively. We conclude that internal resources in the study area should be allocated in a
better manner; the engineering defense capability of the communities graded excellent can be leveraged
in planning, construction, maintenance and management. These strategies will reinforce the
engineering defense capabilities of communities throughout the study area.
**5. Conclusions**
Community disaster prevention and reduction is a basic component of urban disaster prevention
and reduction and plays an important role in the urban security system. In this paper, we constructed an
index system to evaluate the CCDRC; the system includes six primary and 31 secondary indices. We
used entropy-weighted gray target modeling to determine the CCDRC, and GIS spatial overlay analysis
to map the spatial distribution of disaster reduction capability grades. We focused on the SND as a case
study and obtained the following conclusions:
(1) The area's CCDRC is relatively weak; communities with a poor grade account for 63.1% of
the study area, and those with an excellent grade account for only 8.3%. Additionally, the spatial
distribution of CCDRC is uneven. The eastern communities have relatively good capability, while the
western communities adjacent to Lake Taihu have poorer capability. The Shishan Street community,
located in the southeastern part of the study region, has the strongest CCDRC. However, the CCDRC
in the region has a good foundation and it also has large potential for improvement. After analyzing
these results, we propose that CCDRC can be most improved by enhancing their level of public
cognitive ability, rescue and support capacity, and engineering defenses.
(2) In general, the CCDRC of the NDCCDR is at the forefront in the study area, but the CCDRC
among the NDCCDR is not balanced. As shown in Fig.5, among the NDCCDR, the CCDRC of Ylian
is rated "excellent", but Hxiang or Yxin is "good". The CCDRC primary indices of the NDCCDR are
also quite different from each other. As we can see in Fig.6, the public cognitive capability and the
rescue and support capability of the NDCCDR show a general poor feature, but their engineering
defense capabilities and disaster management capability are in the overall advantage. Above
discussions fully indicate that the CCDRC of NDCCDR is not necessarily good. On the contrary, the
evaluation and construction of CCDRC is not only the focus of community disaster prevention and
reduction work, but also the standard for the continuous improvement, construction and promotion of
the NDCCDR. Based on above analysis, we can combine the NDCCDR construction with the CCDRC
construction, from point to face, and ultimately improve the overall level of CCDRC in the region.
This paper takes the municipal area as the case study area, analyzes and compares the CCDRC
and its primary indices between communities in the jurisdiction area. All the pertinent suggestions are
beneficial to the regional functional departments to carry out disaster prevention and mitigation
planning, resource allocation, resident mobilization and administrative decisions within the jurisdiction,
so as not to complicate the implementation of the CCDRC construction due to coordination across
jurisdictions. On the other hand, Suzhou is located in Jiangsu, a strong economic province in China. In
recent years, Gross National Product (GDP) has always been the top of the same level cities. Great
efforts have been made to prevent and reduce disaster (It is evident from the number of CCDRC.).
However, from the analysis results of this paper, the CCDRC of Suzhou is still not satisfactory. It also
shows that our CCDRC building has a long way to go. The overall strengthening of China's CCDRC
will continue for a long time.
The index system is mainly composed of spatial attribute data and non spatial attribute data. We




can obtain the spatial attribute data from a regional geographic database. The non spatial attribute data
can be obtained by means of community disaster monitoring logs, reports and other historical archives,
resident visits and field surveys with the help of local civil affairs department. Therefore, the process of
CCDRC calculation and analysis based on entropy weight - grey target model and GIS overlay method
in this paper, is generally applicable to most of the provinces in mainland China.

As described before, the international community generally follows the model of CBDRM.

Although the model pays more attention to software construction than hardware environment
construction, countries that continue to suffer from various natural disasters are looking for a
manageable community risk management model in recent years, especially in last several years. These
countries (or regions) have a good foundation for strengthening the CCDRC.
The concrete measures contain the use of community-based early warning systems (Paul J. Smith et al.,
2017), community infrastructure exposure risk analysis (S. Fuchs et al., 2015;R. Figueiredo et al., 2016;
Saif Shabou et al., 2017), disaster risk reduction education (Avianto Amri et al., 2017) and
community-level resilience to disaster (Adriana Keating et al., 2017;Estefania Aroca-Jimenez et al.,
2017). The evaluation of CCDRC will help to defend against various kinds of disasters in the
community as a whole, respond at different stages during the development of a disaster, integrate
various resources, and coordinate various disaster reduction approaches. Through the evaluation of
CCDRC, the overall level of CCDRC and the status of the main evaluation indexes can be grasps, so as
to make it easy to take specific measures to effectively strengthen the weak links. The benefits will
offer to communities that are vulnerable to various kinds of disasters, as well as some challenges, such
as different national conditions, unbalanced economic development among countries, great differences
in the system of disaster prevention and reduction, and great differences in the organization level of the
project. To address these challenges, features that make this approach worth considering in the context
of other countries or regions include,

- Besides from the open network GIS platform such as Google earth, many open databases for

disaster prevention and mitigation have been put into use within many countries, and the spatial
information needed for the evaluation of CCDRC can be online obtained. It is convenient to obtain
community attribute information through residents' visit, field investigation and non-governmental
organization's disaster prevention and reduction report under the model of CBDRM.

- The model used in this paper has comparability between spatial units, and has transferability

between regions; using entropy method to determine index weights can avoid the arbitrariness and
unilateralism of subjective weight determination.

- This method is not focused on the index itself, and does not need to establish a function

relationship between the indexes, but rather to model the order relation represented by the  index
value, so it is very easy to operate, and the results of evaluation are diversified.

On the whole, it is world widely feasible to apply the methods introduced in this paper to evaluate

the CCDRC. The research results in this paper can provide a basis for improving the CCDRC and
assist with disaster prevention and reduction strategies. They will also serve as a reference for other
studies of community-level disaster reduction capability. However, due to the large number of indices
in this paper, there must be redundancy between data, so information reduction should be carried out
before evaluation. From the perspective of overall plan of national comprehensive disaster reduction
work, we should prepare to respond at different stages during the development of a disaster. Therefore,
the resilience capability to disaster should also be included in the index system. These two aspects are
the deficiencies of this paper and need to be solved in future research.





Appendix A:
Table A1. Evaluation Index system and the entropy weights of the CCDRC.

| Primary indicator (entropy weight) | Secondary index | Entropy weight | Meaning of index | Data type |
|---|---|---|---|---|
| Disaster risk evaluation capability (0.005) | Number of group event disasters (times/year) | 0.001 | Number of disasters with causalities or property loss caused by grouped events. | |
| | Number of fire disasters (times/year) | 0.001 | Number of disasters caused by the fire (intentionally or unintentionally set). | |
| | Disaster risk | 0.003 | Based on safety and an investigation of vulnerable groups, from 1-4 (weak to strong). | Categorical |
| | Compilation of comprehensive asylum map | 0.045 | Whether the comprehensive asylum map of community is compiled; 1 means "yes", and 0 means "no". | Binary |
| | Disaster reduction capital investment (10,000 RMB /year) | 0.065 | Capital investment specified by the community for disaster prevention and reduction. | |
| Rescue and support capability (0.288) | Number of emergency rescue teams (teams) | 0.023 | Number of emergency rescue teams organized by government, community, and other social organizations. | |
| | Disaster information personnel (persons) | 0.059 | Staff appointed by the community that is responsible for publicizing disaster information. | |
| | Reserve of rescue materials (10,000 RMB) | 0.036 | Converted value of goods and materials allocated or reserved by the community for disaster-related response. | |
| | Per capita medical resources (/10,000 persons) | 0.061 | (Number of medical personnel × Number of beds)/total population of community | |
| | Ratio of non-construction area (%) | 0.006 | The higher the non-building area (such as green areas) ratio in the community, the stronger the buffering capability of disasters and the settlement capability of post-disaster personnel. | |
| | Internal road density (km/km$^2$) | 0.013 | The road length inside the unit area of the community; the higher the density, the more efficient the disaster prevention and emergency response. | |
| Engineering defense capability (0.148) | Standard proportion of building fortifications (%) | 0.001 | The ratio between the planned residential area and the total area of community buildings; the higher the standard fortification ratio, the stronger the residential defense capability. | |
| | Total length of drainage pipeline (km) | 0.030 | Total length of drainage lines (such as rainwater and sewage) and other drainage lines in the community. | |
| | Completion rate of fire protection facilities (%) | 0.099 | Degree of integrity of facilities used for water collection, firefighting and related purposes. | |
| | Area of underground civil defense facilities (m$^2$) | 0.009 | Total construction area of underground residential facilities such as underground car park garages. | |
| Social and economic base support capability (0.012) | Ratio of population from 18-60 years old (%) | 0.009 | Ratio of population aged 18-60 to the total registered household registration population in the community (village). | |
| | Income per capita (10,000 RMB) | 0.003 | Income per capita of the community. | |
| Disaster management capability (0.159) | Daily management system | 0.001 | Whether a performance appraisal system of comprehensive disaster reduction has been established, including institutional measures for the daily management of related personnel and maintenance and management of disaster prevention and reduction facilities. | |
| | Periodic inspection system | 0.003 | Whether hidden dangers are regularly monitored and emergency plans and response for vulnerable populations are reviewed. | Binary |
| | Periodic examination system | 0.004 | Whether comprehensive disaster reduction plans are regularly reviewed, and specific improvement measures are formulated to address the insufficiencies. | |
| | Social mobilization mechanism | 0.007 | Whether a social mobilization mechanism is established. | |
| | Comprehensive disaster reduction archive | 0.114 | Whether a comprehensive disaster reduction archive is established, with archival information such as text and photos that is standard, complete, and easy to consult. | |
| | Comprehensive disaster reduction demonstration community | 0.023 | Whether or not the community is a national comprehensive disaster reduction demonstration community. | |
| | Quality of demonstration community archive | 0.004 | Archive quality for the demonstration community's process of comprehensive disaster reduction (completeness and degree of conformity); 0 indicates "poor", 1 indicates "relatively good", and 2 indicates "good". | Discrete |
| Public cognitive capability (0.381) | Proportion of volunteers (%) | 0.036 | Proportion of volunteers to the total population of the community. | |
| | Frequency of disaster reduction publicity activity (times/year) | 0.130 | Number of publicizing activities carried out every year for disaster prevention and reduction and the number of participants. | |
| | Number of promotional columns | 0.032 | Number of columns publicizing disaster prevention and reduction. | |
| | Number of promotional materials developed (copies) | 0.061 | Number of publicity materials that have been developed (such as popular science books | |





| | | | |
|---|---|---|---|
| | | for disaster prevention and reduction). | |
| Disaster early-warning display screen | 0.035 | Whether the community (village) has a display terminal for disaster early warning (display screen). | Binary |
| Publicizing official account of disaster reduction | 0.067 | Whether there is a WeChat official account for publicizing disaster prevention and reduction (1 indicates "established" and 2 indicates "not established"). | Binary |
| Frequency of emergency practice (times/year) | 0.020 | Number of emergency practice activities organized every year. | |

Note: The indices without a data type annotation are numerical.

*Acknowledgements*: This work was supported by the program " Research for Deconstruction of Types
of Rural concentrated Community and their Evolutional Mechanism in Southern Jiangsu: From the
Perspective of Rural Governance " of the National Natural Science Foundation Program of China
(grant number 51578352).
The data used to create the evaluation indices include (1) spatial data from Suzhou's database,
which were used to calculate indices with ArcGIS software; and (2) attribute data from the Department
of Civil Affairs of Suzhou and statistical analysis by the Civil Affairs Bureau of SND (these data were
obtained from summaries for the various streets).
The authors would like to thank Dong Kaiquan, Xu Shuguang and Wu Pengcheng for their help to
summary the attribute data.

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
