# Peer review of "Evaluating and analyzing the comprehensive community"

_Natural Hazards and Earth System Sciences, 2018_

## Referee Comment (RC1) · Anonymous Referee #1 · 6 Aug 2018

The paper addresses an important and timely topic. Community's role in disaster reduction has been emphasized in both the Sendai Framework for Disaster Risk Reduction (2015-2030) and the Chinese disaster reduction plans. The authors present an index-based approach for calculating the community comprehensive disaster reduction capability (CCDRC) and a case study in Suzhou, China. Such research is definitely needed in the scientific scholarship. Unfortunately, this paper needs to be enhanced thoroughly. I would encourage the authors to revisit the work and ensure that the research is thoroughly backed up by the appropriate literature and is clearly structured and communicated to the readership.

Specific comments: Language: The paper is difficult to understand. The language needs to be improved thoroughly. Particularly, the paper structure needs to be im-

proved.

Literature and significance: The significance of the paper and the proposed index is not clearly explained. The proposed index of the community comprehensive disaster reduction capability (CCDRC) should be presented in a disaster risk management/science context. What is the significance of the CCDRC to the scientific community? How do the findings contribute to relevant literature? How can the results be used to improve disaster risk management? In my opinion, the proposed CCDRC is also highly related to vulnerability and resilience, which have numerous literature including a lot at a community scale. I suggest the authors improve the literature survey and identify the significance of the proposed index and the findings.

Data: Very limited information is given about the data. What are the source, type, reliability, accessibility, and temporal scope of the employed data? What is the spatial size of the study area or the communities?

Methods: The methodology is not well explained. First, the primary and secondary indexes of the CCDRC should be clearly explained. How are the indexes included and structured? Second, what is the relationship between the CCDRC index and the evaluation criteria of the "National demonstration community of comprehensive disaster reduction (NDCCDR)"? What are the reasons that the NDCCDR is imperfect (line 8, 109)? Why the CCDRC and the NDCCDR should be combined (line 25)?

Conclusion: In my opinion, the index-based result can only help to identify relative high or low units but unable to present if the study area is weak or not. I thus disagree with the conclusion that the study area's CCDRC is relatively weak (lines 20–21, 487). It should be confirmed by a comparison between the results of the investigated area with other regions.

---

## Author Comment (AC1) · 10 Aug 2018

First of all, thank the experts for carefully reviewing our papers, putting forward pertinent comments and detailed suggestions. We organized the authors to analyze and discuss the problems raised by experts and formed a consensus. Before the submission of the revised paper, we will retouch the paper word by word, and adjust the structure of the paper according to all the reviews and experts' opinions. We will respond to the above questions one by one.See the PDF file for details.

Please also note the supplement to this comment:
https://www.nat-hazards-earth-syst-sci-discuss.net/nhess-2018-137/nhess-2018-137-AC1-supplement.pdf

[Figure]
First of all, thank the experts for carefully reviewing our papers, putting forward pertinent comments and detailed suggestions. We organized the authors to analyze and discuss the problems raised by experts and formed a consensus. Before the submission of the revised paper, we will retouch the paper word by word, and adjust the structure of the paper according to all the reviews and experts' opinions. We will respond to the above questions one by one.

Literature and significance: The significance of the paper and the proposed index is not clearly explained. The proposed index of the community comprehensive disaster reduction capability (CCDRC) should be presented in a disaster risk management/science context.

*Response: The secondary indices in the index system are described in detail in the appendix 1 at the end of the paper, and the meaning of the primary evaluation indices will be supplemented when submitting the revised manuscript; the first part of the paper (1.Introduction) has made a description of the relationship between CCDRC and disaster risk management. In the revised version, we will strengthen the content in accordance with the experts' suggestions.*

What is the significance of the CCDRC to the scientific community?

*Response: The overall strengthening of regional comprehensive disaster reduction capability was first proposed by the Chinese government in 2006 after several major natural disasters. In 2011, the overall work thought of China was fully elaborated. As the basic component of the urban public disaster prevention and reduction work, the community is the object of the direct action of different types of disaster causing factors, as well as the concrete and direct bearing body of the compound disaster. This has become the common understanding of the international academic circles. The CCDRC quantitative evaluation method and the empirical analysis described in this paper can be used as a universal compliance in the community based disaster risk management model (CBDRM), and it can also be a useful supplement to China's NDCCDR evaluation method.*

How do the findings contribute to relevant literature?

**Fig. 1.**

**Supplement:**

First of all, thank the experts for carefully reviewing our papers, putting forward pertinent comments and detailed suggestions. We organized the authors to analyze and discuss the problems raised by experts and formed a consensus. Before the submission of the revised paper, we will retouch the paper word by word, and adjust the structure of the paper according to all the reviews and experts' opinions. We will respond to the above questions one by one.

Literature and significance: The significance of the paper and the proposed index is not clearly explained. The proposed index of the community comprehensive disaster reduction capability (CCDRC) should be presented in a disaster risk management/science context.

*Response: The secondary indices in the index system are described in detail in the appendix 1 at the end of the paper, and the meaning of the primary evaluation indices will be supplemented when submitting the revised manuscript; the first part of the paper (1.Introduction) has made a description of the relationship between CCDRC and disaster risk management. In the revised version, we will strengthen the content in accordance with the experts' suggestions.*

What is the significance of the CCDRC to the scientific community?

*Response: The overall strengthening of regional comprehensive disaster reduction capability was first proposed by the Chinese government in 2006 after several major natural disasters. In 2011, the overall work thought of China was fully elaborated. As the basic component of the urban public disaster prevention and reduction work, the community is the object of the direct action of different types of disaster causing factors, as well as the concrete and direct bearing body of the compound disaster. This has become the common understanding of the international academic circles. The CCDRC quantitative evaluation method and the empirical analysis described in this paper can be used as a universal compliance in the community based disaster risk management model (CBDRM), and it can also be a useful supplement to China's NDCCDR evaluation method.*

How do the findings contribute to relevant literature?

*Response:1, it can provide reference for disaster risk management technology on community scale; 2. The index system constructed in this paper takes into account the characteristics of all types of disaster risk, the whole process of disaster management and the main body of disaster prevention and disaster preparedness, which can provide decision-making support for the efficient disaster relief management of local governments; 3. The model is simple and easy to operate, and the results of the evaluation are diverse and universally suitable. The results of this paper have a practical reference value for related research and practice in China.*

How can the results be used to improve disaster risk management? In my opinion, the proposed CCDRC is also highly related to vulnerability and resilience, which have numerous literature including a lot at a community scale.

*Response: The evaluation index of CCDRC includes disaster risk assessment, and its evaluation results are helpful to disaster risk management and control. The primary evaluation indices of this paper contains engineering defense capability, which can also reflect the vulnerability, but the CCDRC and resilience also have related characteristics, which is a shortage of this paper and explained at the end of the paper (Line 558-560).*

I suggest the authors improve the literature survey and identify the significance of the proposed index and the findings.

*Response: We will continue to strengthen the literature review, further refine the evaluation indicators, and consolidate the significance of the research results.*

Data: Very limited information is given about the data. What are the source, type, reliability, accessibility, and temporal scope of the employed data? What is the spatial size of the study area or the communities?

*Response: The evaluation index data mainly comes from two aspects, (1) The spatial data is derived from the spatial database of Suzhou in 2015. The file type is mainly shp file. The data is checked before the use of the data. Some new ground objects are updated to keep the time of the attribute data, and some indexes are calculated with the help of the ArcGIS software. (2) The Suzhou Municipal Bureau of Civil Affairs organizes the acquisition of attribute data and entrusts Civil Affairs Bureau of SND to conduct investigation and statistics. All data were reviewed before the summary, so as to guarantee the real and effective data. We will supplement the study area in the revised draft.*

Methods: The methodology is not well explained. First, the primary and secondary indexes of the CCDRC should be clearly explained. How are the indexes included and structured?

*Response: The principle and evaluation steps of the research method were described in the*

*third part. In section 4.3, taking the rescue and support ability of the primary index and the corresponding secondary indices as an example, the calculation process of the primary indices and the secondary indices' target center degree is introduced. Limited to the length of the paper, there is no detailed description of other indices' calculation process.*

Second, what is the relationship between the CCDRC index and the evaluation criteria of the "National demonstration community of comprehensive disaster reduction (NDCCDR)"?
What are the reasons that the NDCCDR is imperfect (line 8, 109)?
Why the CCDRC and the NDCCDR should be combined (line 25)?

*Response: In response to the State Council's spirit of comprehensive strengthening of regional comprehensive disaster reduction capability, the National Disaster Reduction Commission and the Ministry of Civil Affairs have continued to organize and carry out the selection of the "National demonstration community of comprehensive disaster reduction" since 2008. So far, thousands of communities have been selected in China. The NDCCDR evaluation index system is composed of 10 primary indices and 35 secondary indices. However, the system has the following problems, such as redundancy among the primary indices, no indicators that reflect the rescue and security capabilities, and the system has not been updated since 2008; Thus we believe that the evaluation index system of NDCCDR has its own defects. However, NDCCDR plays a demonstration role in the whole country. Local governments take NDCCDR as the basis and direction of the building of CCDRC. Therefore, the evaluation of CCDRC should be carried out in conjunction with NDCCDR.*

Conclusion: In my opinion, the index-based result can only help to identify relative high or low units but unable to present if the study area is weak or not. I thus disagree with the conclusion that the study area's CCDRC is relatively weak (lines 20–21, 487). It should be confirmed by a comparison between the results of the investigated area with other regions.
*Response: Another important reason why we combine CCDRC with NDCCDR is that the CCDRC of NDCCDR is generally better. The findings of this paper also prove this point (Line397-405). In addition, Suzhou City is located in the most developed area of China, with a large investment in disaster prevention and mitigation, and its CCDRC is in the leading position in China as a whole. Based on the above two points, we take the CCDRC of NDCCDR in this area as the criterion to get the conclusion that the CCDRC of the study area is relatively weak. To avoid ambiguity, we still decided to accept the expert's suggestion to replace relatively weak with relatively low (lines 20-21, 487) in our future revised manuscript.*

---

## Referee Comment (RC2) · Anonymous Referee #2 · 28 Sep 2018

The paper presents an interesting overview of a case study to calculate the vulnerability of communities. However, at present the level of presentation in terms of writing and figures needs some work. I outline suggestions for improvement below. Once the presentation issues have been addressed, I would be happy to review the scientific content in further depth.

In the abstract, it is not clear what "National demonstration community of comprehensive disaster reduction (NDCCDR) " is. State that this is a government policy/initiative.

General: the level of English is good, but needs some improvement. For example, line 20 "Despite the capacity to understand and transform nature, human remain vulnerable". "Human" should be corrected to "Humans". I suggest the help of a professional proof reader would help to give the article a final polish.

[Figure]

Line 67 citations such as "Francesco Dottori" should not include first name (also applies to reference list).

Line 74 define SND (although it appears in the abstract, it should be re-defined here).

Introduction in general: the word 'community' is a contested term. Define here what you mean by community. For further info see Cannon, T., 2008.ÂăReducing people's vulnerability to natural hazards communities and resilienceÂă(No. 2008.34). Research paper/UNU-WIDER.

Appendix A table should go into the body text to help the reader understand the calculation of CCDRC

Line 129 unclear what a 'grey target' is. Please explain

Line 279. Give more detail on what the required sample size was and how you met this. What was your sampling strategy? Can you be sure this is representative of the heterogeneity we see within communities?

Figure 4 typo in legend. "Cummunity" should be corrected to "Community". Very difficult to distinguish the line widths for each boundary type. Please use colour or dashed lines to aid visualisation.

Equation 1 and 2 please ensure all variables are defined and clearly explained.

Line 144. GIS overlay is not always an image (which implies a raster dataset) – it could be a vector type dataset. Check throughout this paragraph for use of the word image.

Line 149. Unclear what superposition means within a GIS context. Can you give a reference to the algorithm, or other GIS centered papers that perform this operation.

Line 156 By this point, it is still unclear what the entropy weighting is. Can you give an intuitive definition of this early on in the paper?

Figure 2A Is not easy to visualise. I suggest using colour or reducing the scaling of the

symbols.

Figure 3 Unclear what the hashed areas represent without having to return to the text. Please add a legend and scale bar.

Figure 5 legend is very small. Please increase. I suggest not using so many decimal places as the data you use in calculation does not have this level of precision.

Figure 6 legend has a similar problem to figure 5. I suggest also adding more intuitive labels in addition to the numbers (e.g. 'low', 'medium' 'high').

General: the variables (e.g., public cognitive capability) need further explanation to really understand the results. I suggest presenting each variable with a description in a table fairly early on in the paper.

General: Does non-construction area mean open space? If so, I suggest referring to it in this way.

Figure 8. Unclear what the numbers on the x-axis refer to.

General: for all figures, insure there is a space between the word figure and the number. E.g., Figure 8, not Figure8.

General: there is no real discussion where you bring your findings back into the current literature on this topic. Whereas the conclusions are very long. Conclusions should succinctly summarise the findings and not really introduce new concepts. So perhaps some of this text belongs in the discussion.

---

## Author Comment (AC2) · 6 Nov 2018

The comment was uploaded in the form of a supplement:
https://www.nat-hazards-earth-syst-sci-discuss.net/nhess-2018-137/nhess-2018-137-AC2-supplement.pdf
* * *
The paper addresses an important and timely topic. Community's role in disaster reduction has been emphasized in both the Sendai Framework for Disaster Risk Reduction (2015-2030) and the Chinese disaster reduction plans. The authors present an index-based approach for calculating the community comprehensive disaster reduction capability (CCDRC) and a case study in Suzhou, China. Such research is definitely needed in the scientific scholarship. Unfortunately, this paper needs to be enhanced thoroughly. I would encourage the authors to revisit the work and ensure that the research is thoroughly backed up by the appropriate literature and is clearly structured and communicated to the readership.

Specific comments:

Language: The paper is difficult to understand. The language needs to be improved thoroughly. Particularly, the paper structure needs to be im-proved.

Literature and significance: The significance of the paper and the proposed index is not clearly explained. The proposed index of the community comprehensive disaster reduction capability (CCDRC) should be presented in a disaster risk management/science context.

What is the significance of the CCDRC to the scientific community?

*Response: The overall strengthening of regional comprehensive disaster reduction capability was first proposed by the Chinese government in 2006 after several major natural disasters. In 2011, the overall work thought of China was fully elaborated. As the basic component of the urban public disaster prevention and reduction work, the community is the object of the direct action of different types of disaster causing factors, as well as the concrete and direct bearing body of the compound disaster. This has become the common understanding of the international academic circles. The CCDRC quantitative evaluation method and the empirical analysis described in this paper can be used as a universal compliance in the community based disaster risk management model (CBDRM), and it can also be a useful supplement to China's NDCCDR evaluation method.*

How do the findings contribute to relevant literature?

*Response:1, it can provide reference for disaster risk management technology on community scale; 2. The index system constructed in this paper takes into account the characteristics of all types of disaster risk, the whole process of disaster management and the main body of disaster prevention and disaster preparedness, which can provide decision-making support for the efficient disaster relief management of local governments; 3. The model is simple and easy to operate, and the results of the evaluation are diverse and universally suitable. The*

**Fig. 1.**

**Supplement:**

[revised manuscript text omitted]

In recent years, China has experienced several major disasters, including the Sichuan (Wenchuan) earthquake, the Yushu, Qinghai earthquake, and the Zhouqu debris flows in Gansu Province. Therefore, the concept of comprehensive disaster reduction capability has gained significant traction. It is generally recognized that comprehensive improvements in the disaster reduction capability will reduce or mitigate the causalities and property loss caused by disasters (Hu, 2013). Prior to the International Day for Disaster Reduction in 2006, the State Council of China convened the "Symposium on enhancing the comprehensive disaster reduction capability", which focused on comprehensively strengthening China's comprehensive disaster reduction capability. The concept of comprehensive disaster reduction involves four main aspects: first, preparing for and defending against various kinds of disasters; second, preparing to respond at different stages during the development of a disaster; third, integrating various resources; and fourth, applying various disaster reduction approaches (Lyu, 2011). Many studies have addressed the disaster reduction capability in China and other countries, and their results have been adopted by the governmental organizations and committee and applied in actual regional disaster management situations. These studies have mostly focused on single types of disasters ((Francesco D et al., 2017; Zhang, 2004), single aspects of disaster reduction (Boris F et al., 2016; Daniel Green et al., 2017), and the comprehensive regional disaster reduction capability (Ma, 2007). Relatively few studies have addressed disaster reduction at the community scale. Although some authors have constructed index systems for the ability of communities to prevent and mitigate disasters, they did not propose an in-depth or specific quantitative method (Smith et al., 2017; Yi, 2012). The concrete measures contain the use of community-based early warning systems (Paul J. Smith et al., 2017), community infrastructure exposure risk analysis (S. Fuchs et al., 2015;R. Figueiredo et al., 2016;Saif Shabou et al., 2017), disaster risk reduction education (Avianto Amri et al., 2017) and community-level resilience to disaster (Adriana Keating et al., 2017;Estefania Aroca-Jimenez et al., 2017). However, there is little research on the quantitative evaluation method of CCDRC. In this paper, we aim to address the above situation 
[revised manuscript text omitted]
 (0.005) | Qualitative and quantitative assessment capability by means of hazard identification (including natural and man-made hazards), hazard tracing and frequency recording | Number of group event disasters (times/year) | 0.001 | Number of disasters with causalities or property loss caused by grouped events. | |
| | | Number of fire disasters (times/year) | 0.001 | Number of disasters caused by the fire (intentionally or unintentionally set). | |
| | | Disaster risk | 0.003 | Based on safety and an investigation of vulnerable groups, from 1-4 (weak to strong). | Categorical |
| Rescue and support capability (0.288) | It refers to the ability to deal with emergencies after disasters and to provide materials, equipment and manpower for emergency relief, which is affected by the preparation of plans, communication facilities, material reserves, financial support and rescue teams. | Compilation of comprehensive asylum map | 0.045 | Whether the comprehensive asylum map of community is compiled; 1 means "yes", and 0 means "no". | Binary |
| | | Disaster reduction capital investment (10,000 RMB /year) | 0.065 | Capital investment specified by the community for disaster prevention and reduction. | |
| | | Number of emergency rescue teams (teams) | 0.023 | Number of emergency rescue teams organized by government, community, and other social organizations. | |
| | | Disaster information personnel (persons) | 0.059 | Staff appointed by the community that is responsible for publicizing disaster information. | |
| | | Reserve of rescue materials (10,000 RMB) | 0.036 | Converted value of goods and materials allocated or reserved by the community for disaster-related response. | |
| | | Per capita medical resources (/10,000 persons) | 0.061 | (Number of medical personnel × Number of beds)/total pop | |
| Engineering defense capability (0.148) | It refers to the ability of disaster prevention and mitigation formed by various engineering measures, which is determined by the number, scale and standard grade of disaster prevention projects built to prevent and mitigate disaster occurrence in the region. | Ratio of open space (%) | 0.006 | The higher the open space (such as green areas) ratio in the community, the stronger the buffering capability of disasters and the settlement capability of post-disaster personnel. | |
| | | Internal road density (km/km$^2$) | 0.013 | The road length inside the unit area of the community; the higher the density, the more efficient the disaster prevention and emergency response. | |
| | | Standard proportion of building fortifications (%) | 0.001 | The ratio between the planned residential area and the total area of community buildings; the higher the standard fortification ratio, the stronger the residential defense capability. | |
| | | Total length of drainage pipeline (km) | 0.030 | Total length of drainage lines (such as rainwater and sewage) and other drainage lines in the community. | |
| | | Completion rate of fire protection facilities (%) | 0.099 | Degree of integrity of facilities used for water collection, firefighting and related purposes. | |
| | | Area of underground civil defense facilities (m$^2$) | 0.009 | Total construction area of underground residential facilities such as underground car park garages. | |
| Social and economic base support capability (0.012) | The ability to provide human, financial, resource and environmental support for disaster prevention and mitigation is mainly affected by the level of socio-economic development, the amount of disposable financial revenue and the level of infrastructure development in a region. | Ratio of population from 18-60 years old (%) | 0.009 | Ratio of population aged 18-60 to the total registered household registration population in the community (village). | |
| | | Income per capita (10,000 RMB) | 0.003 | Income per capita of the community. | |
| Disaster management capability (0.159) | It refers to the ability to organize and coordinate various forces reasonably and effectively in order to effectively achieve disaster prevention and mitigation, to formulate reasonable policies, systems and mechanisms, and to flexibly use various methods. It is mainly influenced by such factors as the perfection of the legal system, the ability of social mobilization, the ability of scientific and technological support. | Daily management system | 0.001 | Whether a performance appraisal system of comprehensive disaster reduction has been established, including institutional measures for the daily management of related personnel and maintenance and management of disaster prevention and reduction facilities. | Binary |
| | | Periodic inspection system | 0.003 | Whether hidden dangers are regularly monitored and emergency plans and response for vulnerable populations are reviewed. | |
| | | Periodic examination system | 0.004 | Whether comprehensive disaster reduction plans are regularly reviewed, and specific improvement measures are formulated to address the insufficiencies. | |
| | | Social mobilization mechanism | 0.007 | Whether a social mobilization mechanism is established. | |

| | | Comprehensive disaster reduction archive | 0.114 | Whether a comprehensive disaster reduction archive is established, with archival information such as text and photos that is standard, complete, and easy to consult. | |
| | | Comprehensive disaster reduction demonstration community | 0.023 | Whether or not the community is a national comprehensive disaster reduction demonstration community. | |
| | | Quality of demonstration community archive | 0.004 | Archive quality for the demonstration community's process of comprehensive disaster reduction (completeness and degree of conformity); 0 indicates "poor", 1 indicates "relatively good", and 2 indicates "good". | Discrete |
| Public cognitive capability (0.381) | It refers to the means of raising the public's awareness of disaster prevention and mitigation through traditional propaganda methods such as holding propaganda activities for disaster prevention and mitigation, issuing propaganda materials and first aid drills, installing disaster early warning display screens, setting up publicity signals for disaster reduction and other modern communication technologies, and the proportion of volunteers participating in disaster prevention and mitigation can also be reflected. Public ability in this area. | 
[revised manuscript text omitted]

---

## Author Comment (AC3) · 6 Nov 2018

The comment was uploaded in the form of a supplement:
https://www.nat-hazards-earth-syst-sci-discuss.net/nhess-2018-137/nhess-2018-137-AC3-supplement.pdf
* * *
[Figure]
The paper presents an interesting overview of a case study to calculate the vulnerability of communities. However, at present the level of presentation in terms of writing and figures needs some work. I outline suggestions for improvement below. Once the presentation issues have been addressed, I would be happy to review the scientific content in further depth. In the abstract, it is not clear what "National demonstration community of comprehensive disaster reduction (NDCCDR)" is. State that this is a government policy/initiative.

**Response:** *In lines 8-13 of the abstract, the nature of NDCCDR has been explained, stating that it is a government initiative*

General: the level of English is good, but needs some improvement. For example, line 20 "Despite the capacity to understand and transform nature, human remain vulnerable". "Human" should be corrected to "Humans". I suggest the help of a professional proof reader would help to give the article a final polish.

**Response:** *In the 33 line of the paper, "human" has been modified to "humans".*

Line 67 citations such as "Francesco Dottori" should not include first name (also applies to reference list).

**Response:** *The first name of the 70 and 586 lines has been removed according to the opinion.*

Line 74 define SND (although it appears in the abstract, it should be re-defined here). .

**Response:** *SND has been re-defined in line 82.*

Introduction in general: the word 'community' is a contested term. Define here what you mean by community. For further info see Cannon, T., 2008. ̃a Reducing people's vulnerability to natural hazards communities and resilience ̃a(No. 2008.34). Research paper/UNU-WIDER.

**Response:** *I referred to some of the authoritative Chinese literature and foreign literature. Despite the controversy, we temporarily selected the word community to consistent with China's administrative divisions.*

Appendix A table should go into the body text to help the reader understand the calculation of CCDRC.

**Response:** *Taking Appendix A as part of the body text, it conforms to the logical structure of this article. However, there are more primary indicators and secondary indices. The meaning contents of indices occupy a lot of space. In order to ensure the compactness of the paper, the table is placed in the appendix part. We can consider putting Appendix A table into the body text to help the reader understand the calculation of CCDRC*

Line 129 unclear what a 'grey target' is. Please explain

**Response:** *Grey theory is applied to solve the uncertainty problem under the condition of few data. Grey target theory is a grey relational analysis theory for dealing with pattern sequences. In a set of pattern sequences, the data closest to the propositional scalar value are found to*

Fig. 1.